Journal of Data-centric Machine Learning Research (2026)        Submitted 12/24; Revised 09/25; Published 04/26

# AIDOVECL: AI-generated Dataset of Outpainted Vehicles for Eye-level Classification and Localization

**Amir Kazemi**[1,2] *                                                       KAZEMI2@ILLINOIS.EDU
**Qurat ul ain Fatima**[1]                                                   QFATIMA2@ILLINOIS.EDU
**Volodymyr Kindratenko**[2,3,4]                                             KINDRTNK@ILLINOIS.EDU
**Christopher W. Tessum**[1] *                                              CTESSUM@ILLINOIS.EDU

[1] *The Grainger College of Engineering, Department of Civil and Environmental Engineering, University of Illinois Urbana-Champaign*
[2] *Center for Artificial Intelligence Innovation, National Center for Supercomputing Applications, University of Illinois Urbana-Champaign*
[3] *The Grainger College of Engineering, Siebel School of Computing and Data Science, University of Illinois Urbana-Champaign*
[4] *The Grainger College of Engineering, Department of Electrical and Computer Engineering, University of Illinois Urbana-Champaign*

**Reviewed on OpenReview:** *https://openreview.net/forum?id=59MARvj9vN*

**Editor:** Sergio Escalera

## Abstract

Image labeling is a critical bottleneck in the development of computer vision technologies, often constraining machine learning performance due to the time-intensive nature of manual annotations. This work introduces a novel approach that leverages outpainting to mitigate annotated data scarcity by generating artificial contexts and annotations, significantly reducing labeling efforts. We apply this technique to a particularly acute challenge in autonomous driving, urban planning, and environmental monitoring: the lack of diverse, eye-level vehicle images from desired classes. Our dataset comprises AI-generated vehicle images obtained by detecting and cropping vehicles from manually selected seed images, which are then outpainted onto larger canvases to simulate varied real-world conditions. The outpainted images include detailed annotations, providing high-quality ground truth data. Advanced outpainting techniques and image quality assessments ensure visual fidelity and contextual relevance. Ablation results show that incorporating AIDOVECL improves overall detection performance by up to about 10%, and delivers gains of up to about 40% in settings with greater diversity of context, object scale, and placement, with underrepresented classes achieving up to about 50% higher true positives. AIDOVECL enhances vehicle detection by augmenting real training data and supporting evaluation across diverse scenarios. By demonstrating outpainting as an automatic annotation paradigm, it offers a practical and versatile solution for building fine-grained datasets with reduced labeling effort across multiple machine learning domains. The code and links to datasets are available for further research and replication at `https://github.com/amir-kazemi/aidovecl`.

**Keywords:**   Generative AI, Prompt-Guided Diffusion, Inpainting, Object Detection.

---

∗. Corresponding Authors

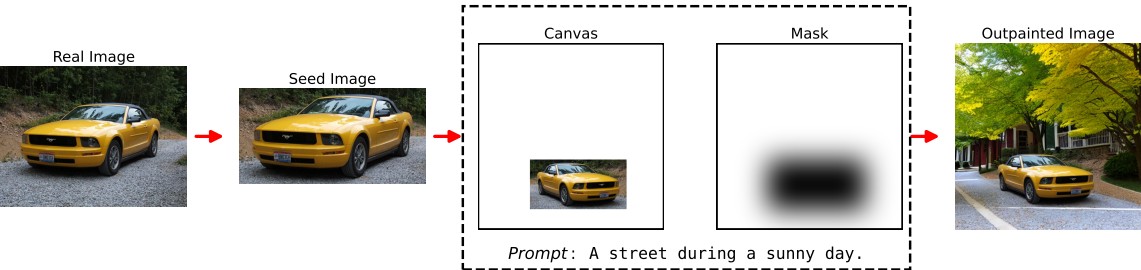

Figure 1: Vehicles from authentic images are randomly scaled and positioned on a canvas, then outpainted using structured prompts and blurred masks.

## 1 Introduction

In recent years, the field of computer vision has undergone significant expansion, leading to transformative changes across various domains such as autonomous driving, urban planning, and environmental monitoring. These advanced technologies may catalyze improvements in transportation, leading to reduced traffic accidents, easing congestion, and mitigating urban air pollution through improved vehicle operations (Dilek and Dener, 2023; Fan et al., 2024; Mavrokapnidis et al., 2021). Central to these advancements is the robustness and accuracy of vehicle classification and localization algorithms, which depend critically on the quality and diversity of the training datasets available. These technologies are continuously being developed to enhance vehicle detection for various applications, ranging from the detection of polluting vehicles (Nisha et al., 2025) to the handling of corner cases in autonomous driving (Lu et al., 2024). Such advancements underscore the critical need for a broad spectrum of vehicle images within specific categories to support these technologies.

Vision-based localization methods, despite their reliance on optimal lighting conditions and susceptibility to obstructions, stand out due to their robust performance in familiar settings. These methods excel by recognizing visual features to enhance navigation and obstacle detection, providing rich contextual data that not only enhances system functionality but also ensures a comprehensive environmental understanding (Lu et al., 2021). Nonetheless, the efficacy of these classification and localization models hinges on access to varied and extensive datasets. Public datasets frequently lack adequate eye-level vehicle representations—essential for autonomous driving and roadside surveillance applications. Furthermore, these datasets often do not include detailed or desired vehicle categorizations, thereby limiting their practical utility. This is further exacerbated by the presence of smaller or partial vehicles in the background which are not annotated. All these shortcomings can adversely affect the performance of vision-based localization machine learning models tasked with operating in dynamic, real-world settings.

Our dataset introduces a novel automatic annotation approach to generating a high-quality collection of AI-generated, eye-level vehicle image datasets; see Figure 1. The methodology commences with the detection of vehicles within manually-selected existing images using a pretrained model. This strategic selection process directly influences the quality and usability of the final dataset, ensuring a comprehensive representation of different

vehicle classes. Upon detection, we crop these images to create "seed images" which can manually be classified as desired. To further augment the dataset and introduce variability, we employ generative AI for outpainting using the seed images. This step involves placing the cropped vehicle images onto larger canvases at random coordinates and scales, thereby enhancing the diversity of the dataset and simulating real-world contexts.

**Contribution** The contributions of this dataset are manifold, significantly enhancing the adaptability and performance of machine learning models. The use of outpainting not only increases the diversity of the dataset but also simulates real-world conditions where vehicles appear at varying scales and locations within an image. This simulation is crucial for developing robust algorithms capable of accurate vehicle classification and localization under diverse operational scenarios. Each image within the dataset is automatically annotated with detailed bounding box coordinates, providing valuable ground truth data for training and evaluation purposes. Also, seed images can readily be categorized into desired classes manually before populating them by outpainting. We may use AIDOVECL as a method to augment real vehicle image datasets which suffer from data scarcity or class imbalance. Our comprehensive ablation study across multiple detection architectures clearly demonstrates the effectiveness of incorporating AIDOVECL, yielding improvements of up to 10% in overall detection performance. The performance gain can rise up to 40% in settings characterized by greater diversity of context, object scale, and placement, with underrepresented classes achieving up to 50% higher true positives. These results highlight AIDOVECL's primary value in augmenting real training data to strengthen vehicle detection, while also serving as a reliable supplement for testing in diverse scenarios. Beyond these performance gains, the study also offers insight into how well existing quality assessment measures capture perceptual and semantic aspects of generated images, particularly in prompt-guided image-to-image tasks like outpainting. By combining advanced detection techniques with state-of-the-art generative models and quality assessment tools, our pipeline and dataset facilitate progress in autonomous driving, traffic analysis, and urban planning. This capability is especially useful in scenarios requiring custom fine-grained classes, reducing labeling effort by decoupling manual classification from object localization and context variation.

The rest of the paper is structured as follows: The **Background** section reviews significant vehicle image datasets, identifying gaps that our approach aims to fill, and also includes an overview of related inpainting and augmentation techniques. The **Methods and Setup** section describes the process of creating seed vehicle images, generating synthetic data through outpainting, and assessing image quality, along with the training and evaluation of the model for vehicle classification and localization. In **Results and Discussion**, we present the outcomes of our image generation and quality assessment, illustrated with examples of outpainted vehicle images, and discuss the performance of the model in classification and localization tasks. Finally, the **Conclusion** summarizes our contributions, highlighting the effectiveness of outpainting as an automatic annotation approach, along with its limitations and possible future work.

## 2 Background

### 2.1 Public Vehicle Image Datasets

Vehicle datasets are essential for improving model accuracy in vehicle recognition in different environments. Despite their importance, these datasets frequently encounter challenges such as limited context around vehicles, broad classifications that obscure finer distinctions, and suboptimal capture angles. These limitations can severely affect tasks requiring detailed environmental information, precise vehicle identification, or particular viewing angles. We briefly review major vehicle datasets to highlight these shortcomings and identify the need for datasets that are more customizable.

The Stanford Cars dataset contains over 16,000 images of 196 classes of cars (Krause et al., 2013), covering a wide range of makes, models, and manufacturing years. This dataset is particularly valued for its high-quality images and annotations, making it a standard benchmark in vehicle recognition and classification tasks. However, many images include unlabeled vehicles in the background, and in some instances, the images do not provide much context, being either too close to the bounding box or having vehicles surrounded by blank spaces, which is not ideal for training highly accurate localization models. Additionally, the dataset does not include heavy trucks and buses.

The Miovision Traffic Camera Dataset (MIO-TCD) (Luo et al., 2018) is another significant resource utilized in vehicle recognition and traffic scene understanding. It comprises a diverse set of traffic-related images designed to facilitate the training and evaluation of various algorithms on mobile platforms. While the dataset offers a broad variety of vehicle types (10 vehicle classes) and traffic scenarios, its major drawback is the lack of near eye-level perspectives. Most images are captured from elevated angles, which can limit the dataset's applicability for applications requiring a pedestrian's, driver's, or roadside point of view.

The Common Objects in Context (COCO) dataset (Lin et al., 2014) is a cornerstone in computer vision, known for its robust features in object detection, segmentation, and captioning across diverse scenes. It excels due to extensive annotations, including precise segmentation masks essential for object localization. However, its vehicle categorization is limited to just three classes: car, bus, and truck. This broad classification can be inadequate for specialized applications needing detailed vehicle recognition, such as advanced safety and surveillance systems.

In summary, datasets like Stanford Cars, MIO-TCD, and COCO advance vehicle recognition but have limitations such as insufficient contextual details, limited vehicle classifications, and non-ideal viewing perspectives, highlighting the need for customizable vehicle datasets. Addressing these gaps is essential for developing application-specific models for evolving vehicle recognition demands.

### 2.2 Inpainting Methods

Image inpainting is a computational technique used to restore and manipulate images, primarily for removing unwanted objects or artifacts. It has applications across various domains, including security, where it plays a critical role in ensuring the integrity of visual information by eliminating potentially compromising elements from shared images. Origi-

nally employed for repairing old or damaged images, it has evolved to address a range of distortions such as text, noise, scratches, and lines. While serving as a valuable tool for enhancing image fidelity, it also presents challenges in forgery detection due to the potential for sophisticated manipulation techniques to conceal traces of editing (Elharrouss et al., 2020).

A variety of strategies are employed to reconstruct missing or corrupted image areas, ranging from progressive (Yu et al., 2018; Sagong et al., 2019; Guo et al., 2021; Zhang et al., 2018a; Li et al., 2020; Yi et al., 2020) and structural information-guided (Nazeri et al., 2019; Yu et al., 2019; Liao et al., 2020; Han et al., 2019; Yang et al., 2020) to attention-based (Zeng et al., 2020; Zhao et al., 2020; Liu et al., 2019; Wan et al., 2021; Yu et al., 2021) approaches. These strategies leverage iterative filling processes, structural and spatial data, and controlled information propagation via masks to mitigate color inconsistencies. Pluralistic approaches address the inherent complexity of inpainting by generating diverse possible reconstructions for each image segment (Xiang et al., 2023). Extending these foundational strategies, various deep learning methods utilize frameworks such as Generative Adversarial Networks (GAN) (Goodfellow et al., 2020) and Variational Autoencoder (VAE) (Kingma and Welling, 2013) integrated with Convolutional Neural Networks (CNN) (LeCun et al., 1995) or Recurrent Neural Networks (RNN) (Graves, 2013) to advance inpainting for a diverse set of applications (Zhang et al., 2023b).

The recent introduction of Latent Diffusion Models (LDM) represents a significant advancement for the inpainting task (Rombach et al., 2022). This method leverages latent diffusion processes to synthesize high-resolution images, focusing on intricate details and structural coherence that outpace traditional models. Unlike previous techniques that often concentrated on replicating textures and patterns, LDMs excel in generating and processing contextual and structural details crucial for complex inpainting tasks. The diffusion process iteratively refines images through transitions across latent representations, enabling effective handling of various distortions without extensive manual parameter tuning. This breakthrough offers a powerful, scalable solution for high-fidelity image inpainting, setting a new benchmark for inpainting tools and markedly enhancing both the quality and realism of inpainted areas in academic and practical applications. Outpainting generates content beyond an image's original boundaries using similar pixel reconstruction techniques as inpainting. Consequently, we employ LDM (Rombach et al., 2022), an advanced method with established pretrained models, to leverage these capabilities.

## 2.3 Related Data Augmentation Methods

Data augmentation is a common technique in machine learning, used to enhance the size and diversity of training datasets, thereby improving the generalizability of models. Traditional methods like flipping and rotation have long been staples, but more sophisticated techniques are necessary to tackle the complex demands of modern AI systems. While augmentation methods such as mosaic and mixup have proved to be promising, the advent of outpainting using diffusion models for object augmentation in diverse environments remains relatively unexplored.

Mosaic data augmentation, introduced in the YOLOv4 paper (Bochkovskiy et al., 2020), represents a significant advancement in training deep learning models for object detection.

This technique involves combining four or more different training images into a single composite image. By doing so, it exposes the model to a more diverse array of object scales, positions, and contexts within a single training example. This approach not only helps in enhancing the detection capabilities of the model, especially for smaller objects, but also significantly boosts its ability to generalize to various real-world scenarios. The inclusion of multiple scenes in one image helps in simulating a more complex and varied visual environment, which is crucial for improving the robustness and accuracy of object detection systems. This method has proven to be especially effective in handling datasets where objects appear in dense configurations and diverse backgrounds.

Mixup data augmentation (Zhang et al., 2018b), significantly enhances the training of neural networks for image classification and related tasks. This method generates a synthetic training sample by combining two images and their corresponding labels using a convex mixture. By doing this, mixup encourages the model to adopt simpler, linear behaviors between training examples, mitigating issues of overfitting and improving model generalization. This linear interpolation between examples provides smoother estimates of uncertainty, which is particularly beneficial in scenarios where models must handle novel or ambiguous inputs. The inherent noise introduced through this process also helps to increase model robustness against small perturbations in input space, thus making it more adaptable and robust in practical applications.

While mosaic and mixup augmentation techniques have made significant strides in enhancing neural network robustness, they also present notable limitations. Mosaic can sometimes result in overly complex images that confuse the model rather than help it learn, especially when objects from different images do not naturally coexist. Similarly, mixup might lead to ambiguous labels if the images combined are too distinct, potentially misleading the model during training. Furthermore, methods like copy-paste (Ghiasi et al., 2021) and x-paste (Zhao et al., 2023), which rely on precise segmentation masks and may not be as useful with just bounding box annotations, often face challenges in integrating objects seamlessly into new scenes, leading to unnatural edges or misplaced objects. In contrast, outpainting using diffusion models emerges as a more promising avenue. These models can generate highly realistic and contextually varied synthetic data by extending the canvas of existing images, creating entirely new scenarios that maintain the integrity and realism of the original content. Additionally, the availability of pretrained prompt-guided diffusion models allows for efficient deployment, significantly speeding up the development cycle. This method not only addresses the limitations of conventional augmentation techniques but also provides a scalable solution to train more robust and adaptable AI systems.

## 3 Methods and Setup

### 3.1 Creating Seed Vehicle Images

**Real Data Collection.** The initial step involves gathering a diverse collection of images of vehicles across the desired categories. This process is manual and entails selecting images where vehicles are presented in a near-eye-level perspective, ensuring that each vehicle is well-separated from smaller background vehicles, if present. Additionally, it is crucial that vehicles are fully visible within the frame, as partial representations result in information loss. For this analysis, our collection comprises over 11,000 vehicle images from various sources

(including Stanford Cars) (Krause et al., 2013; Peng et al., 2019; Addison Howard, 2018; POB, 2023; Hoekstra, 2024a,b; Zatoichi, 2023; Koul, 2024) which are utilized exclusively for academic and non-commercial research purposes.

**Seed Images.** In the next step, images selected during data collection are fed into a selected pretrained vehicle detection model which is able to detect three classes of car, bus, and truck. Note that we use this model to detect the vehicle bounding boxes but not to determine the vehicle class, as we aim to detect different and more classes of vehicles than are available in pretrained models. We elaborate on selecting the detection model in the next paragraph. The largest bounding box detected in each image is then cropped, including a 15% buffer on all sides to enhance the outpainting process. This buffer is crucial as it helps create a smoother transition between the masked area and the newly outpainted regions. Images with vehicle dimensions smaller than 32 pixels in any direction are excluded to ensure high quality and detail in the seed images. Additionally, we retain the uncropped versions of the selected seed images (above 5,000 images) for ablation studies, serving as a baseline. The classification of the seeds is shown in Table 1.

**Model Selection for Detection.** We select the aforementioned detection model by a consensus voting approach, where each model in a predefined ensemble—FCOS (Tian et al., 2019), RetinaNet (Lin et al., 2017), SSD (Liu et al., 2016), MaskRCNN (He et al., 2017), and FasterRCNN (Ren et al., 2016)—detects the largest bounding box for vehicles in a given image. These detections are then compared pairwise using the Intersection over Union (IoU) metric, where each bounding box gains a vote for every other box it significantly overlaps with (as defined by a threshold of 0.95). The model with the highest votes is selected as the primary detector, with the order indicating backup priorities in case the primary model fails. This method leverages multiple models to increase the reliability of the detection by confirming the presence and location of objects through mutual agreement.

Table 1: Vehicle Classification and Subcategories

| ID | Vehicle Class | Subcategories |
|----|---------------|---------------|
| 0 | COUPE | Coupe, Convertible, Cabriolet, or other two-door passenger cars |
| 1 | SEDAN | Sedan, or other four-door passenger cars |
| 2 | SUV | SUV or Crossover |
| 3 | MINIVAN | Minivan or Wagon |
| 4 | MINIBUS | Minibus, Shuttles, or large passenger vans |
| 5 | BUS | City, Coach, Double-Decker, Articulated, or School Bus |
| 6 | VAN | Work, Camper, or Conversion Van |
| 7 | PICKUP | Regular, Crew Cab, or Extended Cab Pickup Truck |
| 8 | TRUCK | Single Unit, Trailer, Articulated, Dump, Tanker, or Mixer Truck |

### 3.2 Synthetic Data Generation

**Canvas Generation and Annotation.** Seed images are randomly scaled and positioned on a $512 \times 512$-pixel blank canvas. The channels of these seed images are also permuted to

diversify the colorization of vehicles. However, this does not alter black and white colors, which is desirable as it maintains the black color of tires. Corresponding annotations are calculated based on the dimensions of the original bounding box (i.e., dividing the buffered seed dimensions by 1.15 to remove the 15% buffer), and the scale and position of the seed on the canvas. We also create a mask image to aid the outpainting modules. This mask maintains the buffer zone—unlike the annotation which accounts for the actual bounding box of the vehicle—and will be blurred on borders to ensure a smooth transition into the outpainted areas.

**Canvas Outpainting.** For the outpainting process, we employ the stable diffusion in-painting model from Hugging Face (SDv1.5, 2025a) which is based on LDM (Rombach et al., 2022). The model utilizes both positive and negative prompts. The positive prompt is constructed dynamically according to the vehicle class and follows the pattern: `"A {location} during {time} with no vehicle."` The `{location}` parameter can take values from { `highway`, `road`, `street`, `downtown`, `driveway`}, tailored to the vehicle type. The `{time}` variable encompasses { `a sunny day`, `a cloudy day`, `evening`, `sunset`, `sunrise`, `dusk`}, providing a temporal context to complement the location. This structured prompt design ensures a contextually appropriate generation of the scene, enhancing the realism of the outpainted areas. To avoid introducing unwanted vehicles, which are not annotated in the previous step, the negative prompt explicitly excludes {`traffic, train, car, truck, bus, van`}. This selection relies on the fact that the masked area remains largely intact during outpainting, allowing us to maintain focus on our vehicle of the desired class. Additionally, we may add {`billboard, text`} to the negative prompt to avoid generating billboard-like artifacts when vehicle branding is treated as background advertising rather than as part of the vehicle itself. Additional terms may be added to the negative prompt to suppress artifacts that compromise the realism and quality of the generated images.

**Image Quality Assessment.** We employ outpainting techniques to extend the boundaries of original vehicle images, creating visually plausible extended scenes. The quality of these outpainted images is rigorously evaluated using three no-reference image quality assessment scores: BRISQUE (Mittal et al., 2012), CLIP-IQA (Wang et al., 2023), and QualiCLIP (Agnolucci et al., 2024). BRISQUE is designed to assess natural scene statistics and quantify potential distortions, typically ranging from 0 to 100, with lower scores indicating better quality. Similarly, CLIP-IQA leverages the CLIP model's ability to assess perceptual quality through deep learning, with scores between 0 and 1 where higher scores suggest superior perceptual quality. QualiCLIP is a self-supervised, opinion-unaware image quality assessment method that fine-tunes the CLIP model to align image representations with quality-related semantics through textual prompts, outputting a score between 0 and 1 where higher values indicate better perceptual quality. We retained images with a maximum BRISQUE score of 22, a minimum CLIP-IQA score of 0.5, and a minimum QualiCLIP score of 0.85, the latter serving as a high semantic quality bar to ensure that only images with strong alignment to high-quality visual semantics were included.

**Data Splitting.** When generating outpainted images for training, validation, and test sets, we ensure that each set is derived from distinct seed images, eliminating any overlap. Specifically, seed images are allocated into separate training, validation, and test groups, ensuring that vehicles in the validation and test sets are not present in the training set.

This approach enhances the robustness of model training by preventing data leakage. Table 2 is presented for further reference throughout the work. To ensure a sufficient and reliable test dataset for evaluation, we allocate 50% of the data to the test split across all studied cases. Meanwhile, the training and validation sets comprise 40% and 10% of the data, respectively.

Table 2: Image distribution by class across different dataset splits

| Split | Size | 0 | 1 | 2 | 3 | 4 | 5 | 6 | 7 | 8 | Background |
|-------|------|-----|-----|----|----|----|-----|----|----|-----|------------|
| Train | 2269 | 15% | 19% | 8% | 4% | 4% | 20% | 3% | 4% | 13% | 10% |
| Val | 562 | 15% | 19% | 8% | 4% | 4% | 20% | 3% | 4% | 13% | 10% |
| Test | 2833 | 15% | 19% | 8% | 4% | 4% | 20% | 3% | 4% | 13% | 10% |

**Background Generation.** To generate background images for our dataset, we utilize the text-to-image stable diffusion model from Hugging Face (SDv1.5, 2025b) based on LDM (Rombach et al., 2022). This process involves selecting from predefined descriptions that vividly depict various urban scenes devoid of any vehicles, such as empty streets and parks at different times and under various atmospheric conditions. These textual prompts guide the generation of detailed and realistic background images while using negative prompts to ensure no vehicles appear. Background images are crucial for improving the performance of visual recognition algorithms by reducing false positives and enhancing the detection of true negatives. By providing clear and unobstructed scenes, the algorithms can more accurately learn what constitutes the absence of target objects, thereby improving overall detection reliability in complex urban environments.

**Pipeline Parameters.** We encourage readers to consult the configuration file provided in the pipeline's repository for the complete list of parameters used throughout the dataset generation process. These parameters—covering both seed image extraction, quality assessment, and outpainting—are primarily determined through iterative experimentation and empirical tuning, and are best understood in the context of the full pipeline setup.

### 3.3 Vehicle Classification and Localization

**Model Training.** In this study, we utilize the YOLOv8 (You Only Look Once) object detection framework (Redmon et al., 2016), specifically employing the YOLOv8 model (Ultralytics, 2024) along with four other previously mentioned models (FCOS, RetinaNet, SSD, and FasterRCNN). We begin by loading pretrained models, which provide a robust starting point due to their weights being trained on a comprehensive dataset. The training process is conducted over 1000 epochs with an early stopping patience of 20 to prevent overfitting to the training data, using a batch size of 4. The learning rates were fixed at $2 \times 10^{-3}$ for FCOS, $4 \times 10^{-3}$ for RetinaNet, $1 \times 10^{-3}$ for SSD, and $8 \times 10^{-3}$ for Faster R-CNN, while YOLOv8 used Ultralytics' default learning-rate configuration—including a starting rate of 0.01, final fraction of 0.01, warm-up over 3 epochs with momentum ramping from 0.8 and bias learning rate of 0.1, SGD momentum of 0.937, weight decay of 0.0005, and a linear learning rate decay schedule with cosine one-cycle scheduling disabled by default.

**Performance Evaluation.** In this work, we compare the performance of trained models using several key metrics: Precision, Recall, mAP50, mAP50-95, Fitness, and F1 Score. Precision measures the accuracy of positive predictions, indicating the proportion of correct positive identifications out of all positive predictions made. Recall evaluates the model's ability to identify all relevant instances, representing the fraction of true positives detected among the actual positives. The mAP50 evaluates the model's accuracy at a 50% IoU threshold, assessing how well the predicted bounding boxes align with the ground truth. The mAP50-95 extends this evaluation across IoU thresholds from 50% to 95%, providing a robust measure of model performance under varying levels of detection stringency. Fitness is specifically defined as a weighted sum of metrics: 10% from mAP@0.5 and 90% from mAP@0.5:0.95, omitting weights for Precision and Recall by default. Finally, the F1 Score combines Precision and Recall into a single metric that quantifies the model's overall accuracy and completeness in detection tasks. In addition to overall metrics, we evaluate class-wise detection performance using confusion matrices and qualitative analysis of confused, false negative, and false positive instances.

### 3.4 Compute Resources

Our experiments were conducted on the Delta advanced computing and data resource at the National Center for Supercomputing Applications (NCSA). The specific node utilized features 4 CPU cores, each allocated with 16 GB of memory, totaling 64 GB, and is equipped with an NVIDIA A100-SXM4-40GB GPU. The compute time for outpainting each $512 \times 512$-pixel image is approximately five seconds, and generating an image satisfying quality scores requires $30 \sim 60$ seconds on average.

## 4 Results and Discussion

### 4.1 Comparing Detection Models for Seed Image Extraction

The vehicle seed images, designated for outpainting, are identified from a manually classified dataset of over 11,000 images using established object detection models. The selection of models for this task is determined by a consensus voting mechanism that considers pairs of models with an IoU exceeding 0.95, as previously described. Figure 2 displays the consensus level of each model, calculated as the percentage of affirmative votes received from other models, averaged across all vehicle images. For example, a consensus level of 0.5 indicates that the model agrees with two out of four possible models on average. As shown in the figure, FCOS receives the most approval, followed by RetinaNet, SSD, MaskRCNN, and FasterRCNN, reflecting their relative consensus rankings. We prioritize models for seed extraction in the specified order, using subsequent models as backups in the rare case that a model fails to detect an object.

Although this comparative study initially aimed to identify the most approved model within our pipeline, the resulting radar chart offers key insights for advancing future object detection models. For vehicle detection, the focus of this study, SSD, MaskRCNN, and FasterRCNN are less effective in bounding box extraction compared to FCOS and RetinaNet. Moreover, all models show a notably lower consensus level for trucks relative to other vehicle classes. This disparity is likely due to several reasons: frequent model failures

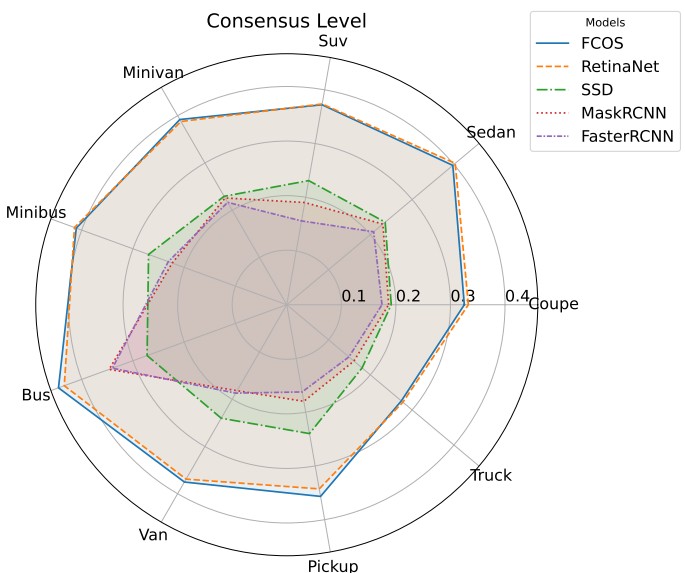

Figure 2:  Consensus level of every detection model is defined as the percentage of affirmative votes received from other models, averaged across all objects in that class.

in detecting trucks, possibly stemming from inadequate training set diversity. Additionally, the subjective nature of manual truck annotations in training data, where the trailer part might be inconsistently included, and the presence of advertisements on trucks, which complicates detection during inference, are significant challenges. To mitigate these issues, future detection models must be trained on more diverse datasets and ensure uniform and comprehensive annotation practices, particularly for complex vehicle categories like trucks.

## 4.2  Outpainted Vehicles

Samples of outpainted vehicles, as generated using the methodology described in the previous section, are illustrated in Figure 3. The diverse set of vehicle images results from varying the prompts for location and time of the scene, alongside the inherent randomness in generative inpainting models, and the seed's scaling and positioning on the canvas. Additionally, employing no-reference image quality assessments, such as QualiCLIP, BRISQUE, and CLIP-IQA scores, ensures a degree of realism and natural appearance within the generated images. Across repeated attempts for each seed image, the method finds an outpainted image whose context is more relevant to the type and lighting of the vehicle by meeting the quality criteria.

The scene integrity across outpainted images underscores the capability of the inpainting model to generate relevant context around a vehicle, viewed at street level or generally from an eye-level perspective. Notably, the use of negative prompts to exclude unwanted vehicles has proven successful in almost all cases. This approach suggests that explicitly mentioning the masked object as a positive prompt is unnecessary, as the entire image generation is

conditioned upon it. Although it might seem counterintuitive, this approach helps us avoid background vehicles that are not annotated automatically.

To quantify both the visual and semantic quality of the generated images, we report the distribution of two sets of qualities in Figure 4: 1) The same scores we used to vet the outpainted images—QualiCLIP, CLIP-IQA, and BRISQUE—satisfy the imposed thresholds of 0.85, 0.5, and 22, respectively. Moreover, mean QualiCLIP and CLIP-IQA scores are at least 0.90, with BRISQUE averaging around 13, all indicating high visual quality. 2) The sentence similarity of prompts for generation and AI-generated captions of outpainted images using existing pretrained models. For the latter, using the sentence transformer, we compare the sentence similarity between `"A {vehicle class} on or in a {location} during {time}"` as the expected caption and the captions predicted by BLIP and ViT-GPT2, yielding scores in [-1,1], where a score closer to 1 indicates high semantic similarity and a score closer to -1 indicates opposing semantics. In both cases, the average sentence similarity is approximately 0.5, indicating a significant and meaningful degree of semantic alignment between the generation prompts and the AI-generated captions. We additionally compare the sentence similarity between BLIP and ViT-GPT2, as this reveals how consistently the semantics of a generated image are captured across different captioning models, providing an extra proxy for cross-model semantic fidelity. The relatively high mean similarity of 0.7 indicates that the generated images contain semantically coherent content, as shown by consistent interpretations from distinct captioning models, which are less likely to hallucinate in different directions when the input is contextually grounded.

## 4.3 Overall Detection Performance Across Models

We conducted an ablation study that spans five modern object-detection architectures (FCOS, RetinaNet, SSD-Lite, FasterRCNN, and YOLOv8). For every model we evaluate four train/test configurations—Real/Real, Aug./Real, Real/Aug., and Aug./Aug.—to isolate the benefit of introducing synthetic data during training and the robustness to distribution shifts at inference. Here, augmentation signifies that each real image is accompanied by a synthetic counterpart from AIDOVECL; the generation pipeline surpasses a 91% success rate against our quality criteria after up to 100 iterations per image. Within every configuration we further vary on-the-fly augmentations—namely the mixup ratio and mosaic probability—to probe how aggressive blend-and-compose policies interact with the underlying data composition. Six standard metrics (Precision, Recall, F1, mAP50, mAP50-95, and Fitness) are reported for all combinations.

Part of the results in Tables 3, 4, and 5 compare a baseline scenario—where each detector is both trained and evaluated on real images—to a scenario where the training dataset is augmented with AIDOVECL, while evaluation remains solely on real images. That is, we first focus on "Real/Real" and "Aug./Real" results. Table 3 demonstrates that adding synthetic AIDOVECL images improves almost all metrics when no additional on-the-fly augmentations are applied. Improvements of up to 10% are noted across each model. Increasing mixup to 0.5 as shown in Table 4, however, does not lead to consistent or significant improvements when training data is augmented. This suggests that pixel-level blending from mixup effectively reduces overfitting. Consequently, if inference data closely matches training distribution, substantial benefits from AIDOVECL may be limited when using mixup

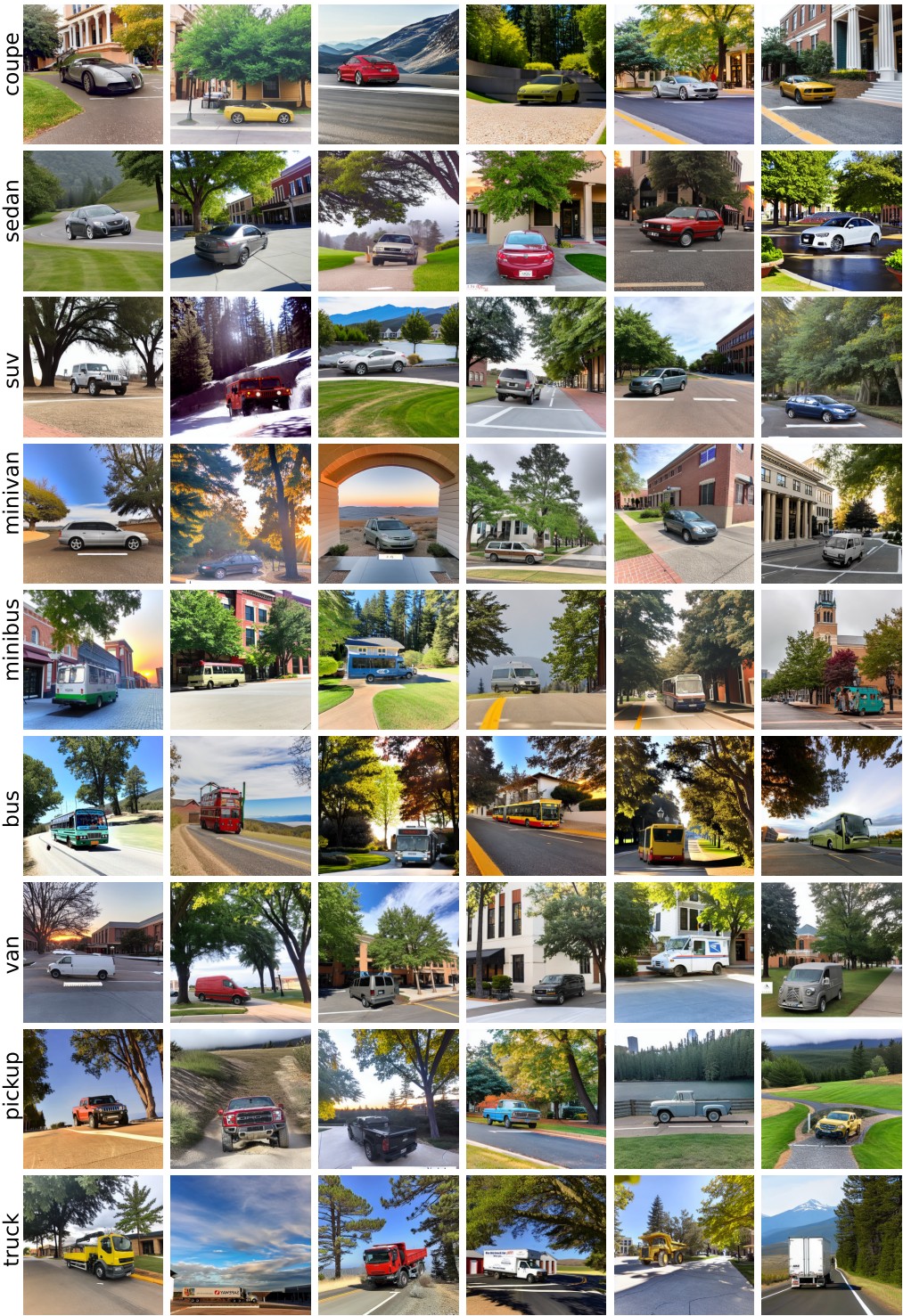

Figure 3: Outpainted images of various vehicle classes satisfying criteria for BRISQUE, CLIP-IQA, and QualiCLIP.

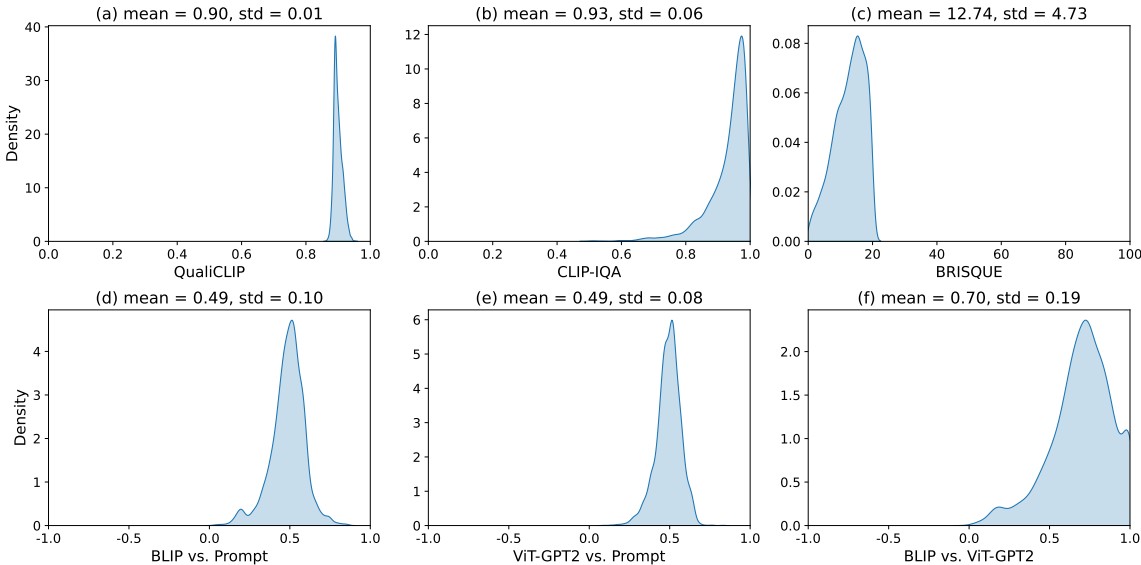

Figure 4: KDE plots for visual quality scores (a–c) and semantic similarities of prompt and captions (d–f). Larger values generally indicate better quality and semantics, except for BRISQUE where smaller values are better.

simultaneously. On the other hand, Table 5 shows that when only mosaic augmentation is applied with a probability of 0.5, there is an improvement of up to about 10%. This indicates that mosaic augmentation alone does not diminish the advantages offered by our augmentation. Therefore, when inference remains within the distribution, AIDOVECL augmentation provides clear performance benefits for most cases.

Comparing the results for "Aug./Aug." and "Real/Aug." highlights the importance of augmenting the training data with AIDOVECL. Across all models, configurations, and metrics presented in Tables 3, 4, and 5, "Real/Aug." consistently underperforms compared to "Aug./Aug.". Although augmenting with AIDOVECL does not introduce entirely new vehicle objects, it instead rescales and relocates existing vehicles into new backgrounds; therefore, it remains crucial for the model to encounter these backgrounds during training to mitigate out-of-distribution inference issues. As demonstrated in all cases across the tables, "Aug./Aug." significantly outperforms "Real/Aug." by up to around 40%, indicating that augmenting training data with AIDOVECL effectively prepares the models for encountering vehicles in contexts different from those present in the real training dataset.

Our comprehensive ablation study across multiple detection architectures clearly demonstrates the effectiveness and necessity of incorporating AIDOVECL-generated synthetic data during training. Synthetic augmentation notably enhances performance, with improvements of up to about 10%. Importantly, synthetic augmentation substantially boosts robustness against distribution shifts at inference, achieving performance gains of up to around 40% when evaluation contexts differ from training. Thus, augmenting real datasets with AIDOVECL significantly strengthens model reliability and generalization in varied deployment scenarios.

| Model | Train/Test | Metrics | | | | | |
|---|---|---|---|---|---|---|---|
| | | Precision | Recall | F1 Score | mAP50 | mAP50-95 | Fitness |
| FCOS | Real / Real | 0.63 | 0.94 | 0.75 | 0.63 | 0.42 | 0.44 |
| | Aug. / Real | **0.65** | **0.97** | **0.78** | **0.65** | **0.44** | **0.46** |
| | Real / Aug. | 0.56 | 0.90 | 0.69 | 0.56 | 0.41 | 0.42 |
| | Aug. / Aug. | **0.69** | **0.98** | **0.81** | **0.69** | **0.53** | **0.55** |
| RetinaNet | Real / Real | 0.60 | 0.79 | 0.68 | 0.60 | 0.45 | 0.46 |
| | Aug. / Real | **0.61** | **0.81** | **0.70** | **0.61** | 0.45 | **0.47** |
| | Real / Aug. | 0.53 | 0.74 | 0.62 | 0.53 | 0.40 | 0.42 |
| | Aug. / Aug. | **0.67** | **0.84** | **0.74** | **0.67** | **0.54** | **0.55** |
| SSD | Real / Real | 0.74 | 0.83 | 0.78 | 0.74 | 0.64 | 0.65 |
| | Aug. / Real | **0.77** | **0.85** | **0.81** | **0.77** | **0.65** | **0.67** |
| | Real / Aug. | 0.55 | 0.70 | 0.62 | 0.55 | 0.45 | 0.46 |
| | Aug. / Aug. | **0.74** | **0.84** | **0.79** | **0.74** | **0.62** | **0.64** |
| FasterRCNN | Real / Real | 0.76 | 0.92 | 0.83 | 0.76 | 0.57 | 0.59 |
| | Aug. / Real | **0.77** | **0.94** | **0.84** | **0.77** | **0.58** | **0.60** |
| | Real / Aug. | 0.66 | 0.86 | 0.75 | 0.66 | 0.50 | 0.52 |
| | Aug. / Aug. | **0.78** | **0.93** | **0.85** | **0.78** | **0.63** | **0.65** |
| YOLOv8 | Real / Real | 0.81 | 0.83 | 0.82 | 0.86 | 0.81 | 0.81 |
| | Aug. / Real | **0.88** | **0.86** | **0.87** | **0.91** | **0.87** | **0.87** |
| | Real / Aug. | 0.79 | 0.79 | 0.79 | 0.82 | 0.76 | 0.77 |
| | Aug. / Aug. | **0.87** | **0.85** | **0.86** | **0.90** | **0.86** | **0.87** |

Table 3: Evaluation metrics with on-the-fly data augmentation during training configured as mixup=0, mosaic=0.

## 4.4 Class-wise Performance of YOLOv8 Detection

In this section, we examine detection performance at the class level, unlike the previous section which focused on overall dataset metrics. We use the YOLOv8 model, known for its strong balance between accuracy and computational efficiency, to analyze confusion matrices and visual examples of false positives, false negatives, and class confusions. This class-wise view helps us better understand where the model struggles and how augmentation with AIDOVECL leads to improvements, particularly for rare, visually similar, or previously hard-to-detect classes. By illustrating representative samples, we analyze which mismatches disappear, arise, or endure when the training data is augmented with AIDOVECL—hereafter referred to as resolved, emergent, and persistent mismatches.

**Confusion Matrices.** Figures 5 and 6 show confusion matrices for the YOLOv8 model tested on real and augmented data, respectively. In each case, the model is trained on both real and augmented datasets under various on-the-fly augmentation configurations. Overall, confusion, false negatives, and false positives generally decrease across classes when the model is trained on augmented data compared to real data, and this improvement becomes even more pronounced when testing is also conducted on augmented data. Exposure to

| Model | Train/Test | Metrics | | | | | |
|---|---|---|---|---|---|---|---|
| | | Precision | Recall | F1 Score | mAP50 | mAP50-95 | Fitness |
| FCOS | Real / Real | **0.71** | **0.98** | **0.83** | **0.71** | 0.47 | 0.50 |
| | Aug. / Real | 0.70 | 0.97 | 0.82 | 0.70 | **0.49** | **0.51** |
| | Real / Aug. | 0.65 | 0.98 | 0.78 | 0.65 | 0.46 | 0.48 |
| | Aug. / Aug. | **0.76** | 0.98 | **0.86** | **0.76** | **0.61** | **0.63** |
| RetinaNet | Real / Real | **0.73** | **0.91** | **0.81** | **0.73** | **0.56** | **0.58** |
| | Aug. / Real | 0.71 | 0.88 | 0.79 | 0.71 | 0.52 | 0.54 |
| | Real / Aug. | 0.62 | 0.83 | 0.71 | 0.62 | 0.48 | 0.49 |
| | Aug. / Aug. | **0.76** | **0.90** | **0.82** | **0.76** | **0.61** | **0.63** |
| SSD | Real / Real | **0.74** | **0.88** | **0.80** | **0.74** | **0.61** | **0.62** |
| | Aug. / Real | 0.71 | 0.83 | 0.77 | 0.71 | 0.59 | 0.61 |
| | Real / Aug. | 0.54 | 0.73 | 0.62 | 0.54 | 0.43 | 0.44 |
| | Aug. / Aug. | **0.70** | **0.83** | **0.76** | **0.70** | **0.58** | **0.59** |
| FasterRCNN | Real / Real | 0.79 | 0.98 | 0.88 | 0.79 | 0.54 | 0.57 |
| | Aug. / Real | **0.80** | 0.98 | 0.88 | **0.80** | **0.57** | **0.60** |
| | Real / Aug. | 0.69 | 0.93 | 0.79 | 0.69 | 0.50 | 0.52 |
| | Aug. / Aug. | **0.81** | **0.98** | **0.88** | **0.81** | **0.62** | **0.64** |
| YOLOv8 | Real / Real | 0.82 | 0.83 | 0.83 | 0.87 | 0.82 | 0.83 |
| | Aug. / Real | **0.86** | **0.86** | **0.86** | **0.90** | **0.86** | **0.87** |
| | Real / Aug. | 0.80 | 0.81 | 0.80 | 0.84 | 0.79 | 0.79 |
| | Aug. / Aug. | **0.86** | **0.85** | **0.86** | **0.90** | **0.86** | **0.86** |

Table 4: Evaluation metrics with on-the-fly data augmentation during training configured as mixup=0.5, mosaic=0.

vehicles in varying sizes and contextual backgrounds during training enhances the model's ability to detect them during testing, as it learns to generalize across spatial scales and diverse scenes. In over 80% of class-configuration combinations, the use of AIDOVECL leads to an increase in true positive counts. The most substantial gain—approximately a 50% increase in true positives—is observed for the underrepresented van class, likely because such classes have more room for performance improvement compared to already well-represented ones. Notable reductions in confusion include fewer misclassifications of vans as minivans or minibuses, coupes and SUVs as sedans, and a decrease in mutual confusion between minibuses and buses, with a general decline in mutual confusion among similar vehicle types. For other classes, improvement or deterioration in confusion is relatively minor. AIDOVECL helps to reduce overall false negatives and false positives in almost all ablation cases, because it increases the frequency of object appearances in diverse contexts, making the model less likely to miss true objects and less prone to trigger detections on background patterns.

| Model | Train/Test | Metrics | | | | | |
|---|---|---|---|---|---|---|---|
| | | Precision | Recall | F1 Score | mAP50 | mAP50-95 | Fitness |
| FCOS | Real / Real | 0.61 | 0.94 | 0.74 | 0.61 | 0.41 | 0.43 |
| | Aug. / Real | **0.67** | **0.95** | **0.78** | **0.67** | **0.47** | **0.49** |
| | Real / Aug. | 0.66 | 0.95 | 0.78 | 0.66 | 0.50 | 0.52 |
| | Aug. / Aug. | **0.74** | 0.95 | **0.83** | **0.74** | **0.60** | **0.61** |
| RetinaNet | Real / Real | 0.70 | 0.84 | 0.76 | 0.70 | 0.54 | 0.56 |
| | Aug. / Real | **0.74** | **0.90** | **0.81** | **0.74** | **0.56** | **0.58** |
| | Real / Aug. | 0.72 | 0.86 | 0.78 | 0.72 | 0.59 | 0.60 |
| | Aug. / Aug. | **0.78** | **0.92** | **0.84** | **0.78** | **0.65** | **0.66** |
| SSD | Real / Real | 0.71 | 0.81 | 0.75 | 0.71 | **0.61** | 0.62 |
| | Aug. / Real | **0.72** | **0.83** | **0.77** | **0.72** | 0.60 | 0.62 |
| | Real / Aug. | 0.68 | 0.80 | 0.74 | 0.68 | 0.57 | 0.58 |
| | Aug. / Aug. | **0.72** | **0.83** | **0.77** | **0.72** | **0.62** | **0.63** |
| FasterRCNN | Real / Real | 0.68 | 0.92 | 0.78 | 0.68 | 0.47 | 0.49 |
| | Aug. / Real | **0.69** | **0.96** | **0.80** | **0.69** | 0.47 | 0.49 |
| | Real / Aug. | 0.68 | 0.90 | 0.77 | 0.68 | 0.51 | 0.53 |
| | Aug. / Aug. | **0.75** | **0.95** | **0.84** | **0.75** | **0.58** | **0.59** |
| YOLOv8 | Real / Real | 0.83 | 0.81 | 0.82 | 0.86 | 0.81 | 0.82 |
| | Aug. / Real | **0.84** | **0.87** | **0.85** | **0.90** | **0.86** | **0.86** |
| | Real / Aug. | 0.81 | 0.81 | 0.81 | 0.85 | 0.79 | 0.80 |
| | Aug. / Aug. | **0.85** | **0.85** | **0.85** | **0.89** | **0.85** | **0.86** |

Table 5: Evaluation metrics with on-the-fly data augmentation during training configured as mixup=0, mosaic=0.5.

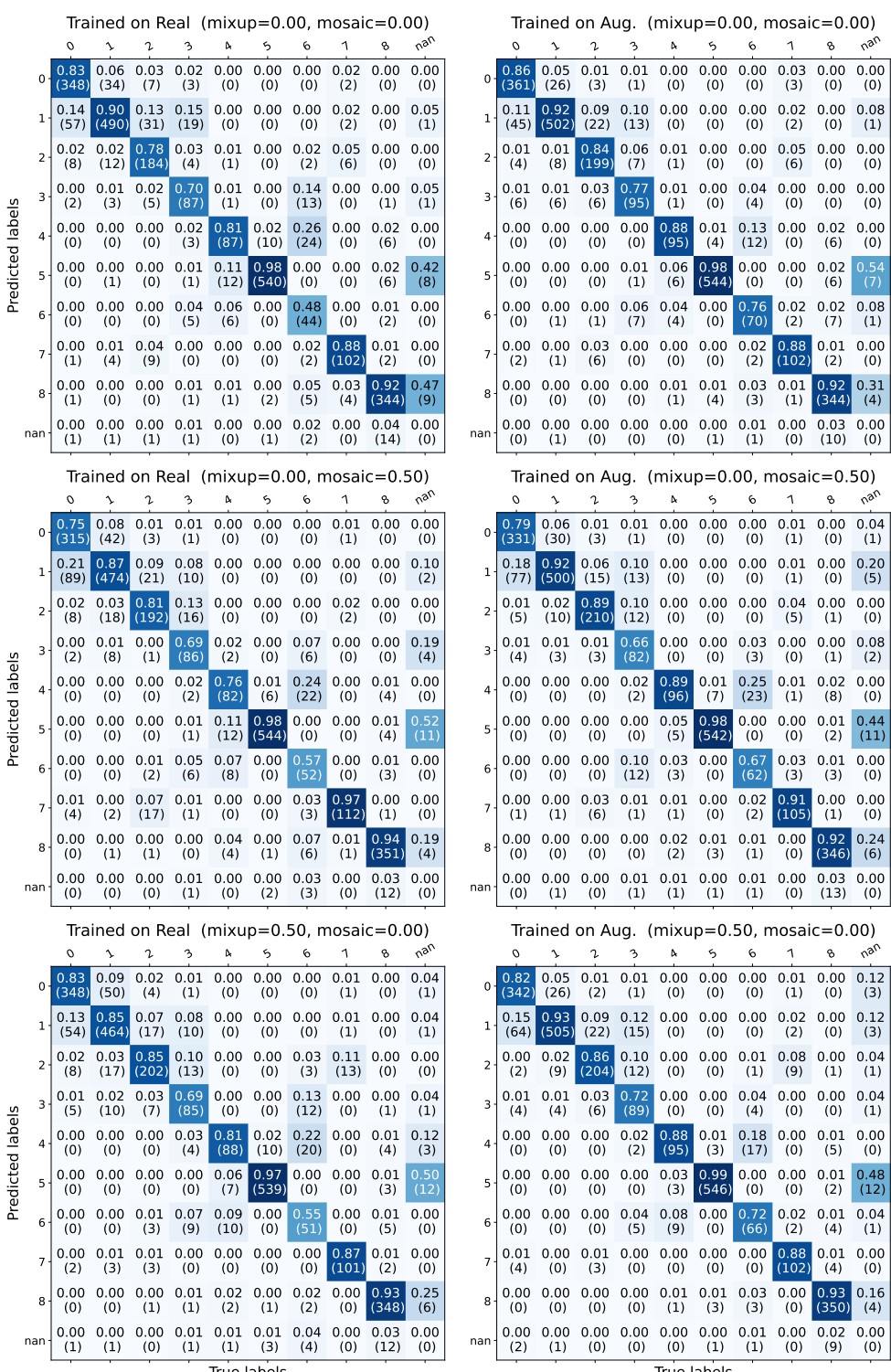

Figure 5: Confusion matrices of the YOLOv8 model tested on the real dataset under different augmentation settings, including dataset augmentation with AIDOVECL and on-the-fly mixup/mosaic.

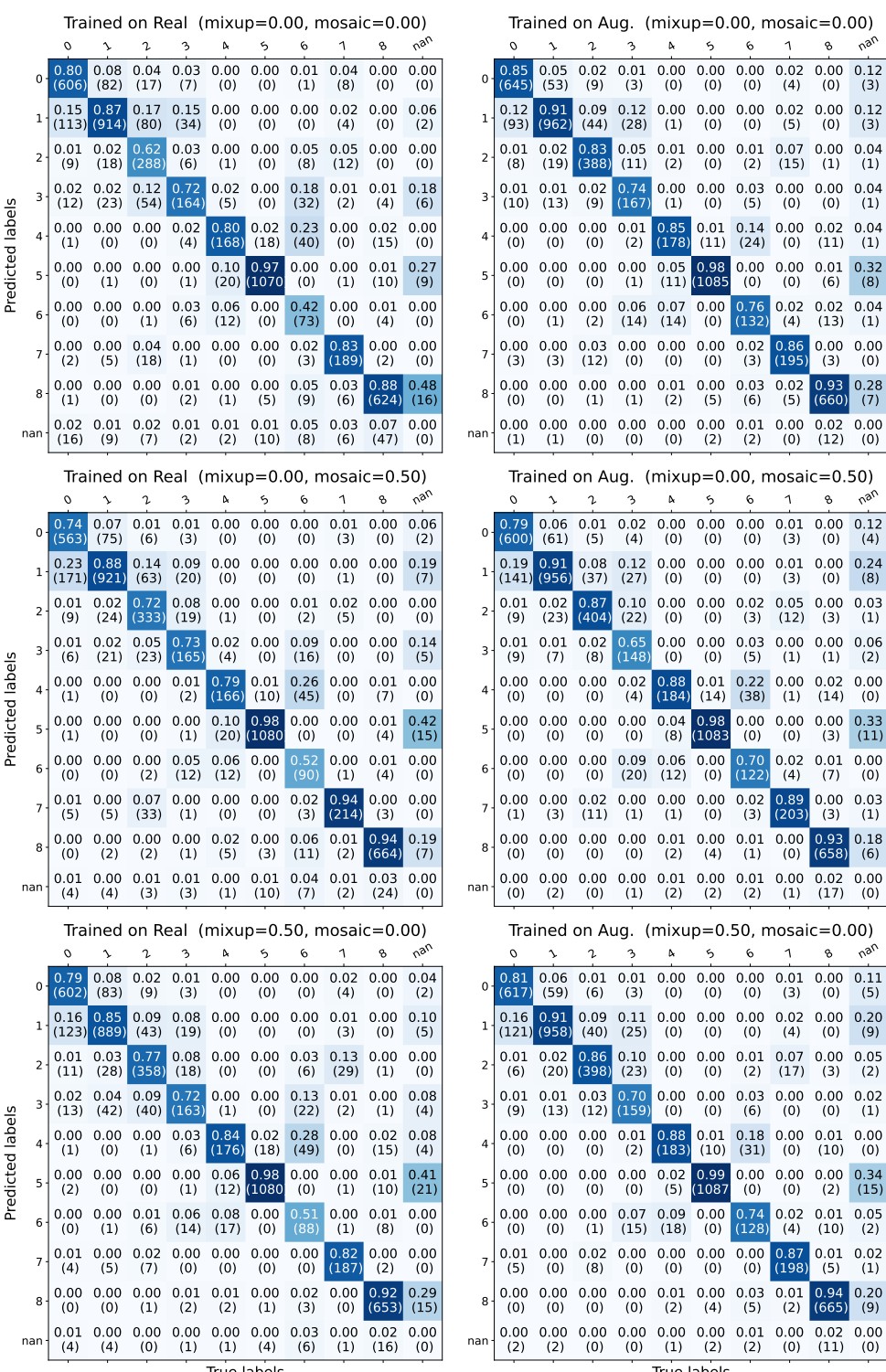

Figure 6: Confusion matrices of the YOLOv8 model tested on the augmented dataset under different augmentation settings, including dataset augmentation with AIDOVECL and on-the-fly mixup/mosaic.

**Confused Instances.** Confused instances are those in which the predicted bounding box closely matches the ground truth location, but the predicted class label does not. Figure 7 presents examples of such instances across the previously defined categories—resolved, emergent, and persistent—after training with the AIDOVECL-augmented dataset and testing on real data without mixup or mosaic augmentations. Notably, the number of emergent confused instances is roughly half that of the resolved ones, which aligns with the trends observed in the confusion matrices. Interestingly, at least three emergent examples—specifically, f1, e2 and e3—demonstrate more accurate bounding box predictions than the corresponding ground truth annotations, despite the class confusion. This highlights the model's ability—when trained on augmented data—to localize objects precisely even when class prediction becomes challenging.

**False Negative Instances.** Occasionally, no prediction is made for a ground truth object, or the predicted bounding box—though class-correct—falls below the IoU threshold; such cases are considered false negatives. As shown in Figure 8, these instances are more often resolved than newly introduced when training with augmented data. Most false negatives involve vehicles that are partially occluded, visually faded due to distance or weather, or have unusual body shapes not seen during training. In some persistent cases, the YOLOv8 model correctly detects vehicles that are missing from the ground truth due to annotation constraints—each image is labeled with only the largest bounding box, omitting smaller vehicles as in a5, d5, b8, and d8. Though rare, such examples demonstrate the model's ability to detect background vehicles missed by PyTorch detectors, particularly when trained on augmented data (e.g., compare a5 vs. d5, where Torch detected a person as a vehicle by mistake). In other cases—often involving articulated trucks—YOLOv8 even outperforms the ground truth annotations by Torch, so the resulting false negative count underestimates actual performance.

**False Positive Instances.** The YOLOv8 model may also produce false positives by detecting vehicles for which no corresponding ground truth exists. Such cases often stem from inaccurate, partial, or missing annotations, and more rarely from background elements—such as buildings—being mistakenly detected as vehicles. As shown in the persistent samples of Figure 9, smaller vehicles (typically buses) omitted from annotations, or partially labeled trucks, are successfully detected by the YOLOv8 model. Notably, training with augmented data is more likely to yield accurate bounding boxes in such cases (e.g., compare a7 vs. d7 and b7 vs. e7). Instances of accurate detection of partially annotated vehicles may appear in both false negative and false positive categories because the predicted box, though correct, may not sufficiently overlap with the ground truth due to the annotation's smaller size or limited extent. When the IoU falls below the matching threshold, this results in a false negative from the ground truth's perspective and a false positive from the model's, as neither considers the other an acceptable match.

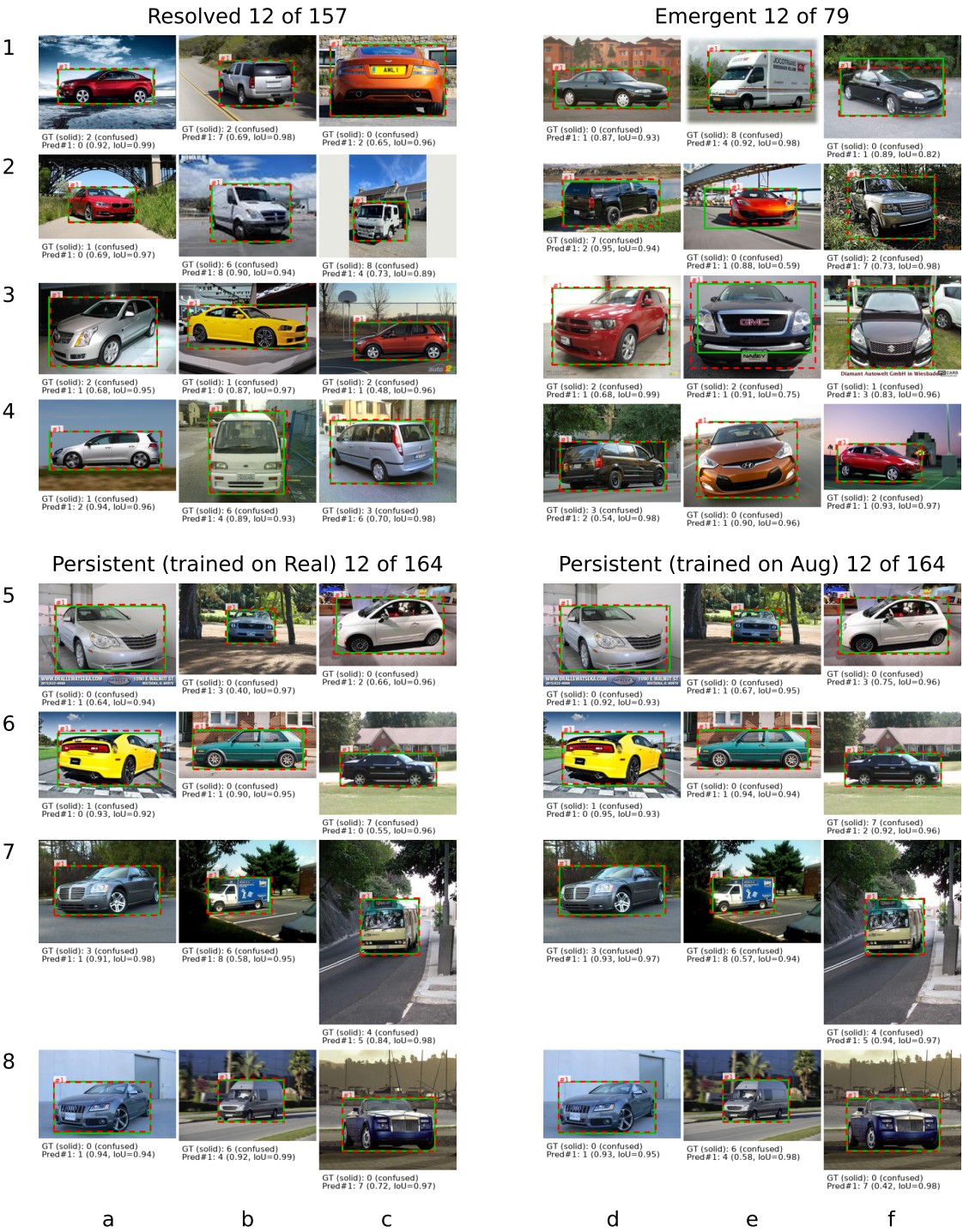

Figure 7: Confused instances, where the ground truth is correctly localized but misclassified by the prediction, shown as GT (solid): [class] (confused) and Pred#[n]: [class] (confidence, IoU).

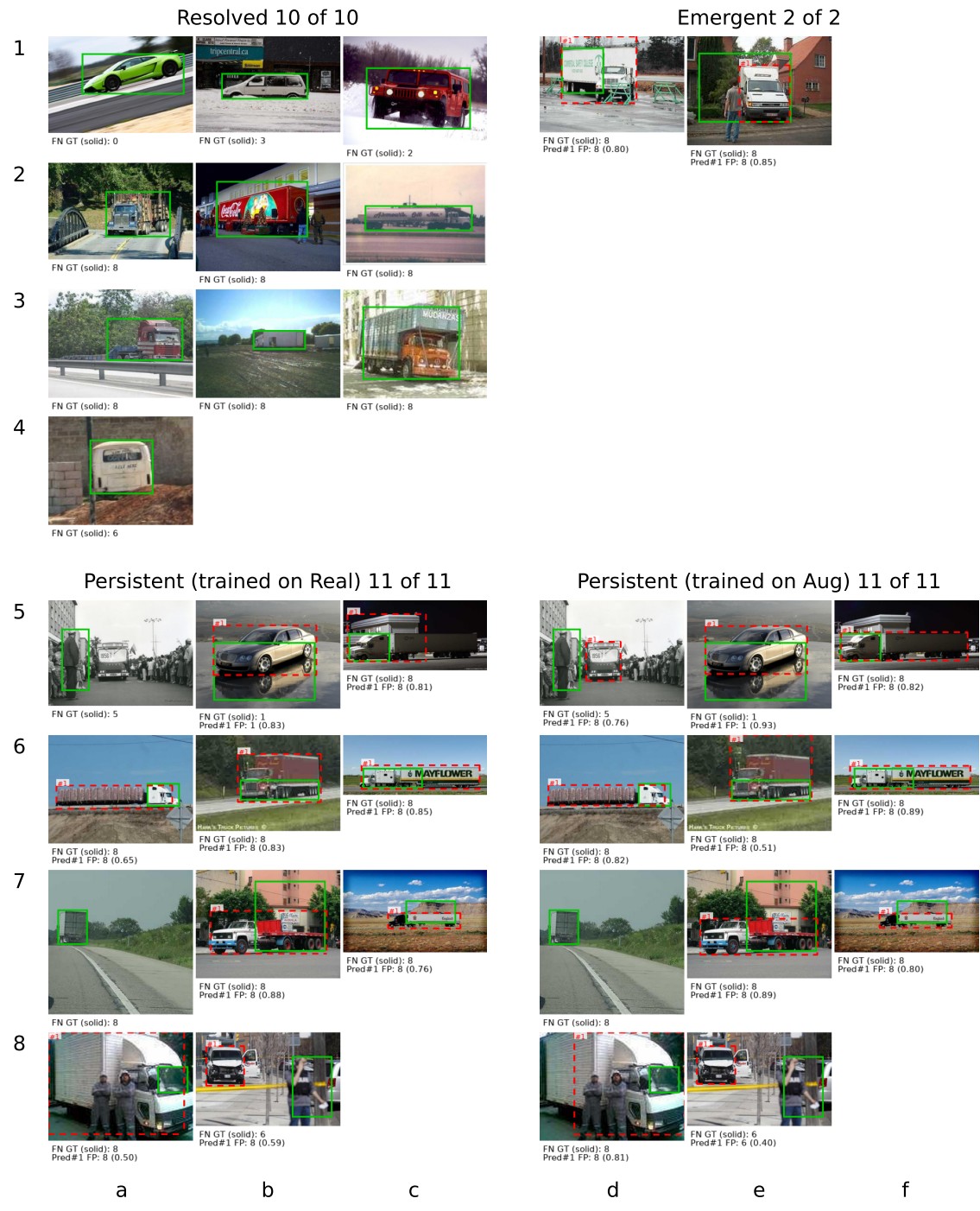

Figure 8: False negative instances, where the ground truth is either not detected or only partially matched with an IoU below the threshold, shown as GT (solid): [class] and Pred#[n] FP: [class] (confidence).

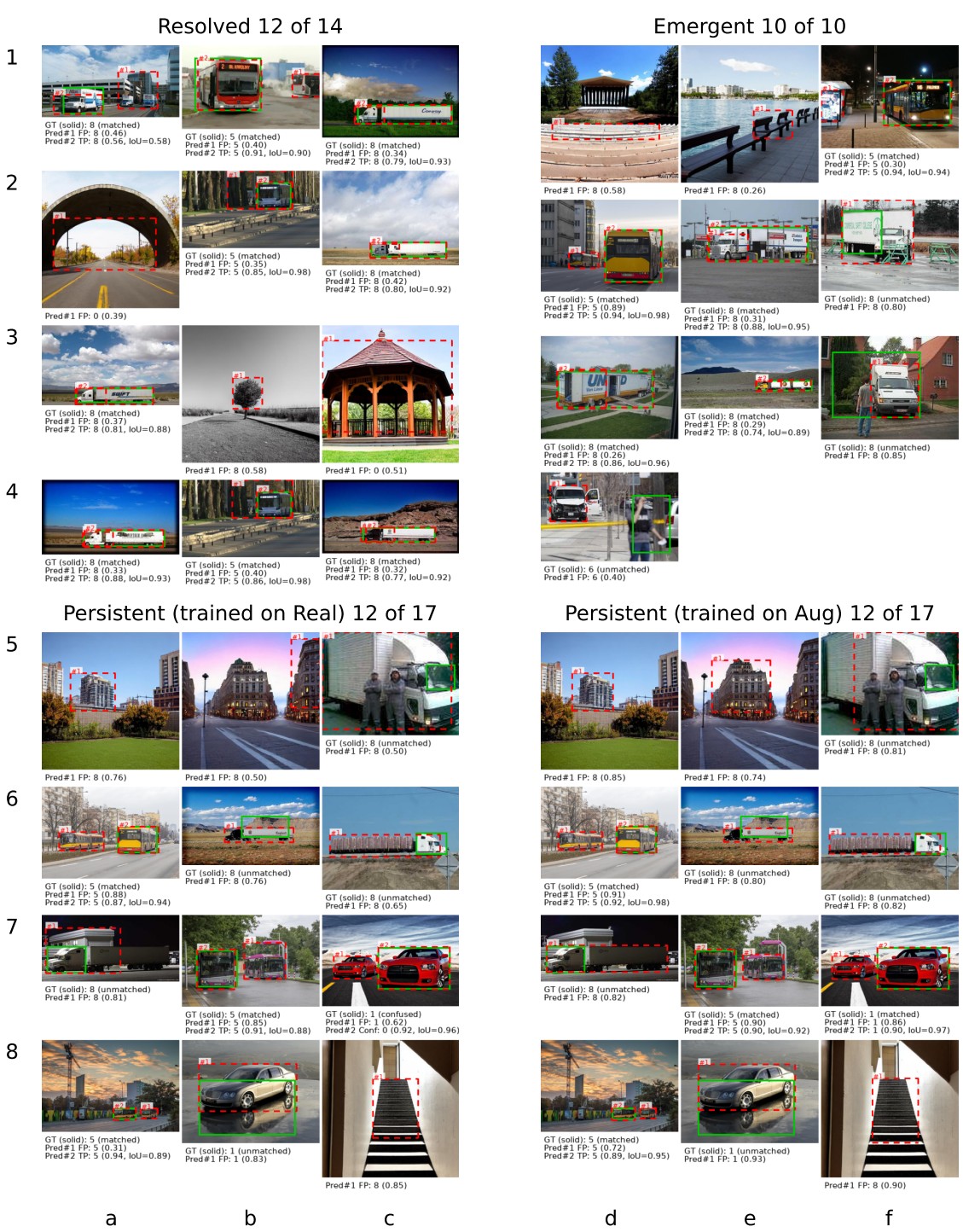

Figure 9: False positive instances, where a prediction corresponds to no or only partially matched ground truth, shown as GT (solid): (matched/unmatched/confused) [class] and Pred#[n] (FP/TP/Conf): [class] (confidence, IoU).

## 5 Limitations

Our reliance on pretrained detection and inpainting models introduces certain limitations, particularly when dealing with non-standard or atypical vehicles that may not be recognized during seed extraction or may result in unrealistic outpainting. These limitations are most relevant when the goal is to include classes not well represented in the pretrained models' training data. However, this does not undermine our claims, as our focus—reflected throughout the paper and in the use of the term *vehicle*—has been on standard, commonly encountered types. Importantly, the proposed pipeline remains flexible: additional seed images or new object classes can be easily incorporated to enhance robustness within existing categories or to support training models that target broader or more specialized vehicle types.

When trying to outpaint multiple objects on the same canvas, the model faces several challenges. With only general scene cues (such as location or timing), it must guess each object's placement and relation to the scene from mask geometry and surrounding pixels. In the LDM architecture, the entire masked area is encoded and denoised as a single spatially coherent latent feature map, with no mechanism to treat disconnected regions independently. If these masked areas are handled in a single pass, the model processes them as one editing region, leading to cross-attention competition—where one object dominates and others may suffer from omissions or partial presence (i.e., objects that are faint or blended into the background, making them no longer clearly distinct). This one-pass complexity forces the model to resolve all spatial, style, and detail constraints simultaneously. At the same time, it must maintain global coherence—such as consistent lighting, shadows, and perspective—while ensuring smooth background blending at multiple transition boundaries, making errors more likely than when outpainting a single object. The problem is further amplified if the pretrained model has rarely or never seen such objects co-occurring in its training data. Although not a replacement for real multi-object training data, techniques like mixup and mosaic can help fill the gap by encouraging generalization to more complex, composite, or partially occluded scenes. Moreover, this limitation becomes less critical when AIDOVECL is seen as part of a broader augmentation pipeline rather than a standalone dataset.

Despite strict visual and perceptual vetting using quality assessment tools, manual inspection reveals that approximately 5.5% of the outpainted images still exhibit various artifacts. See the affected outpainted images in the validation split of the data in Figure 10. Heavy vehicles such as trucks and buses may suffer from non-smooth transition boundaries, patchwork effects, or irrelevant backgrounds, although these artifacts are rare. The main culprit is that the pretrained LDM was not trained specifically on vehicle-centric scenes, especially not on less frequent categories like trucks and buses. Unwanted vehicles, which are not automatically annotated, may still appear despite negative prompts including different vehicle types, likely due to their co-occurrence with other elements in the training data of the LDM. Also, we do not have any mechanism to rule out distorted roads or seasonal mismatches between regions of the image. Defective vehicles may also appear, as detection models for seed vehicle extraction may not fully capture vehicles—typically trailers. These rare cases remain usable for the downstream task of detection and are retained to benchmark our augmentation pipeline, as they err on the side of caution by potentially underestimating

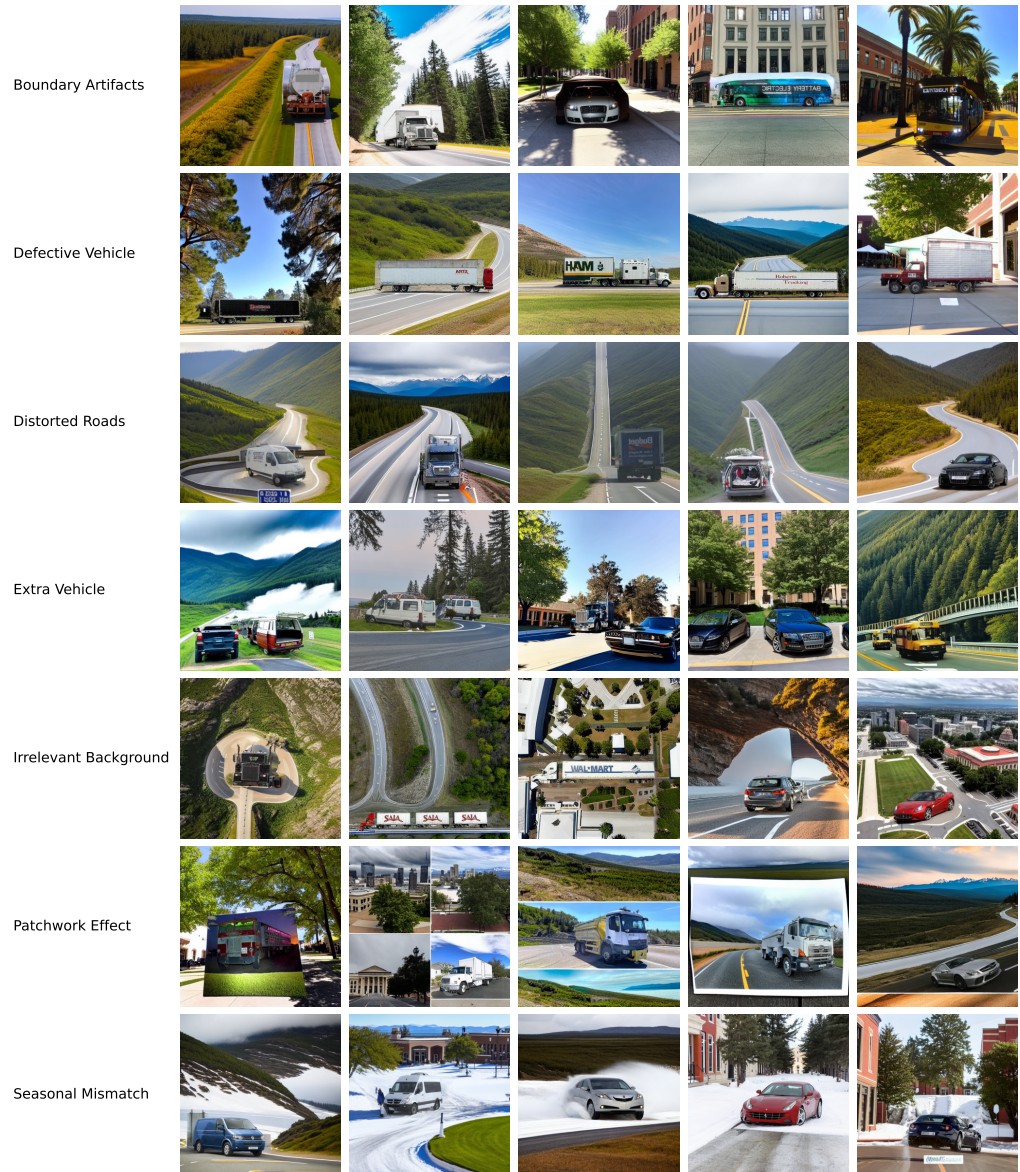

Figure 10: Artifacts are found in 5.5% of outpainted vehicles, despite strict quality checks. These cases are manually excluded from the curated dataset for aesthetic and realism.

our performance. However, they are excluded from the curated dataset to ensure full realism and aesthetic quality, as noted in Data Availability and User Framework.

## 6 Future Work

Active learning offers a promising direction for facilitating efficient seed pool construction in AIDOVECL by prioritizing which unlabeled vehicle crops—initially unclassified and manually assigned to custom categories—should be classified first. In a cold-start setting, diversity-first selection through core-set methods can ensure that the earliest labeled vehicle crops are representative of the broader data distribution (Sener and Savarese, 2017). Task-agnostic latent-space selection, as in variational adversarial active learning, can further identify underrepresented or heterogeneous vehicle types before a classifier is trained (Sinha et al., 2019). Once a small classifier exists, uncertainty-based querying using Bayesian methods can target ambiguous or confusing vehicle crops (Gal et al., 2017), while batch acquisition strategies such as BADGE can balance informativeness and diversity when expanding seeds in groups (Ash et al., 2019). In addition, loss prediction modules can highlight samples expected to be difficult for the classifier (Yoo and Kweon, 2019), ensuring annotation effort is concentrated where it yields the most improvement. Finally, self-training approaches can automatically assign labels to high-confidence samples while leaving manual effort for low-confidence or rare vehicle types (Xie et al., 2020). Together, these approaches could reduce manual labeling costs and accelerate seed pool growth, thereby strengthening the foundations of AIDOVECL.

To overcome the limitations of single-pass multi-object outpainting—where disconnected regions compete for attention and global coherence is difficult to maintain—several promising strategies can be drawn from recent advances in conditional and controllable diffusion models: One approach is to guide generation with the original image (anchor view) and location-aware features (relative positional embeddings) (Zhang et al., 2024). Alternatively, a grounded outpainting model inspired by GLIGEN (Li et al., 2023) could condition on inputs defining the canvas layout to preserve object positions while generating coherent context. Structural inputs such as depth or edge maps, used in ControlNet (Zhang et al., 2023a), can guide outpainted regions around multiple objects to follow scene geometry and blend naturally into the environment. MultiDiffusion (Bar-Tal et al., 2023) can be adapted for multi-object outpainting by coordinating generation across background and object-adjacent regions so that lighting, perspective, and style remain consistent. Composable diffusion methods (Liu et al., 2022) can improve multi-object handling by combining independent condition scores to reduce concept interference. Attend-and-Excite (Xu et al., 2018), while originally developed for text-to-image generation, can inspire methods that keep attention balanced across multiple objects, helping preserve coherence in their expanded surroundings. All these methods require fine-tuning of existing diffusion models to incorporate the additional conditioning and control mechanisms they rely on. Moreover, advancing data curation with ML to predict plausible object–background pairings can further enhance realism and ensure synthetic data better reflects real-world variability.

Future work may explore strategies to further reduce the occurrence of outpainting artifacts. Fine-tuning the pretrained LDM on vehicle-centric datasets—including underrepresented classes such as trucks and buses—could improve generation fidelity for those categories. Exploring state-of-the-art scoring mechanisms, such as artifact detectors, might help identify and discard images with non-smooth transitions, patchwork effects, or irrelevant backgrounds more effectively. Additionally, refining prompt engineering techniques or

applying context-aware negative prompting may help suppress unwanted vehicles, distorted roads, seasonal mismatches, and ambiguous objects more reliably. Overall, while most generated images are usable for detection tasks, improving visual realism and scene consistency remains an important direction.

## 7 Conclusion

The AIDOVECL dataset represents a significant advancement in the field of machine learning for vehicle classification and localization. It addresses the critical issue of annotated data scarcity in desired classes and perspectives, and mitigates the problem of unlabeled background objects in public datasets. By incorporating generative AI techniques—specifically prompt-guided outpainting—AIDOVECL not only enriches the diversity of eye-level vehicle images but also effectively simulates realistic urban traffic scenarios. This approach is crucial for training algorithms capable of performing accurately in dynamic real-world environments. AIDOVECL's automatic annotation, which potentially mitigates the need for manual annotation, offers a substantial improvement as demonstrated across various detection models, with or without on-the-fly augmentation methods. It can augment real datasets with class imbalances, thus enhancing the adaptability and performance of machine learning models. The dataset and its associated pipeline meet immediate training needs and foster future advancements in computer vision technologies for applications ranging from autonomous driving to urban planning.

## Dataset Availability and User Framework

The AIDOVECL resource is released as both (i) a dataset, comprising curated and benchmark versions, and (ii) a modular framework for generating customized augmented datasets. The dataset is publicly hosted on Hugging Face, with an accompanying dataset card describing its structure, splits, and annotation format, and is distributed under the `CC-BY-4.0` license, and is archived under DOI `10.57967/hf/8444` (Kazemi et al., 2026). The associated generation pipeline, including source code and usage instructions, is publicly available on GitHub under the `MIT` license:

```
https://huggingface.co/datasets/amir-kazemi/aidovecl
https://github.com/amir-kazemi/aidovecl
```

We provide both a curated dataset—combining real and AI-generated samples in a conventional split ratio for computer vision tasks, with rare cases containing artifacts excluded through manual inspection—and a research archive of benchmark datasets for reproducing the paper's results, which adopt a higher test split ratio to enable more rigorous evaluation. The latter is organized as follows: Raw unvetted images are provided under `real_raw` (with class-wise subfolders for buses, coupes, minibuses, minivans, pickups, sedans, SUVs, trucks, and vans). From these, vetted seed images are extracted and stored under `real`, which contains `images/`, `labels/`, and a `real.yaml` configuration file for YOLO-style training. Augmented data produced by AIDOVECL is placed under `augmented` along with `real` ones in a corresponding directory, with a parallel organization of `images/`, `labels/`, and `augmented.yaml`.

The pipeline is intended not only as a static dataset release but also as a tool for researchers to generate additional augmented data tailored to their own target domains. Users are encouraged to apply the framework to broaden vehicle classes, adapt to novel environments, or scale dataset size as needed. In such cases, they are required to cite the present AIDOVECL work as the originating framework. While we do not plan to expand the official dataset with additional vehicle categories at this stage, we will strive to address reported bugs and ensure that the repositories remain usable and well-documented. In this way, the dataset and framework are both stable and extensible, supporting community-driven evolution.

## Broader Impact Statement

In this research, we introduce a method for generating training datasets for vehicle detection and classification using outpainting techniques. The potential benefits of our work include enhanced accuracy in automated vehicle detection, which could improve urban planning, traffic management, and autonomous driving, contributing to safer and more efficient urban environments. While our dataset is carefully curated to focus on vehicle features that support these applications, we acknowledge that any form of data capture involving vehicles may contain elements that could raise privacy concerns. We emphasize the importance of ethical guidelines and responsible use of technologies to mitigate potential risks associated with data misuse. In particular, AIDOVECL is intended primarily for research and educational purposes, not for surveillance or other uses that could infringe on individual rights, especially since it includes labels only at the vehicle-class level rather than finer-grained vehicle attributes, thereby limiting its utility for such purposes. Users of the dataset and framework are expected to comply with relevant legal regulations and institutional review requirements, and to exercise caution when extending the pipeline to new domains. To support transparency, the dataset (CC-BY-4.0) and code (MIT) licenses are explicitly provided in their respective repositories.

## Acknowledgments and Disclosure of Funding

The authors acknowledge support from the University of Illinois NCSA Faculty Fellowship Program, the Zhejiang University and University of Illinois Dynamic Research Enterprise for Multidisciplinary Engineering Sciences (DREMES), and the University of Illinois Discovery Partners Institute (DPI). This research used the Delta advanced computing and data resource which is supported by the National Science Foundation (award OAC 2005572) and the State of Illinois. Delta is a joint effort of the University of Illinois Urbana-Champaign and its NCSA (Gropp et al., 2023).

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
