# OpenReview forum: "AIDOVECL: AI-generated Dataset of Outpainted Vehicles for Eye-level Classification and Localization"
_DMLR — Accepted by DMLR_

### Review · Reviewer_4zBi · 2025-02-07

**Recommendation:** 2
**Confidence:** 2

**Summary Of Contributions:**

This study introduces a novel outpainting-based approach to generate artificial contexts and annotations, reducing the need for manual labeling. The method is applied to challenges in autonomous driving, urban planning, and environmental monitoring, where diverse, eye-level vehicle images are often scarce. The dataset consists of AI-generated vehicle images, created by detecting and cropping vehicles from seed images, then outpainting them onto larger canvases to simulate varied real-world conditions. These outpainted images come with detailed annotations, providing high-quality ground truth data. Advanced outpainting techniques ensure visual fidelity and contextual relevance.

**Strengths:**

A new and useful dataset generated by AI models which is beneficial to the research community

**Audience:**

Yes

**Claims And Evidence:**

nO

**Datasets And Benchmarks:**

Yes

**Extended Submissions:**

n/a

**Limitations:**

No experiments illustrating the quality of the dataset

**Requested Changes:**

- Benchmark existing models on the test data.
- Add experiments by training models on the training data and test them.

**Strengths And Weaknesses:**

Strengths:

- The idea of augmenting current dataset of outpainted vehicles is interesting. It could mitigate the problem of lack of enough data points for classification and localization
- The authors release their code and generated dataset, which is beneficial to the community.
- Synthetic data generation is becoming increasingly important and exploring this idea in vehicles data generation is a good starting point.

Weaknesses:

- This work is merely a data augmentation task with a rather limited scope.
- I am concerned about the quality of the generated data. For example, in Figure 3, looking at the third and sixth images in the pickup row, the generated vehicles appear unclear and unnatural. The last two images in the truck row also exhibit similar issues. This raises concerns about the dataset’s practicality. This is actually my biggest concern. I also carefully check the released data, and I find a large portion of unnatural pictures.
- The usefulness of this dataset is not very clear. If it is intended as a test set, it should benchmark the performance of existing models. If it is meant to be a training set, it should be used for model training and evaluated on publicly available test sets.

---

### Review · Reviewer_S8zT · 2025-03-01

**Recommendation:** 2
**Confidence:** 2

**Summary Of Contributions:**

This paper studies an interesting problem: outpainting vehicles for automatic annotations and labeling based on generative AI. The vehicles come from real images and are slightly modified before being outpainted according to some prompts. Experiments show that the model trained with augmented data achieves better performance, especially for the underrepresented classes.

**Strengths:**

See above.

**Audience:**

Yes

**Claims And Evidence:**

Yes

**Datasets And Benchmarks:**

Yes

**Extended Submissions:**

N/A

**Limitations:**

I am mostly concerned with the image quality of the new datasets.

**Requested Changes:**

- Additional experiments: compare the results on the the real-world image dataset before and after training with the new proposed dataset, to check if the new dataset will harm the performance on real-world images. Also, compare the real-world performance with the model trained on a real-world similar dataset and the proposed dataset to see if the new dataset can help the model perform better than the real-world dataset.

**Strengths And Weaknesses:**

The problem in this paper is well-motivated and I believe generative AI can be a good option to solve it. The pipeline in this paper makes sense to me and the paper is well-written. The weaknesses mainly lie in the concern about the quality of the generated images. More specifically,

- In the outpainting process, stable diffusion models are adopted given a combination of texts. However, it is hard to verify if the text itself makes sense to humans and if the image content aligns well with the text, which cannot be reflected by quality assessment because it is more on the semantic and reasoning level.

- Speaking of the image quality,  the evaluation criteria seem inaccurate, meaning some of the outpainted images shown in the paper look unrealistic to humans. The images in Fig. 3 tend to have colorful backgrounds and the snow in Fig. 1 is everywhere on the trees but not on the grass, which is not reasonable. To make the dataset more convincing, I recommend conducting extra experiments to compare with real-world images.

- Regarding the seed images, I wonder what if the pre-trained models cannot detect bounding boxes if the bound boxes are too rare and out-of-distribution. Maybe large vision models can perform better to detect more bounding boxes.

---

### Review · Reviewer_zb9g · 2025-08-07

**Recommendation:** 2
**Confidence:** 2

**Summary Of Contributions:**

Paper proposes generative AI formed dataset and related tools to implement the generation pipeline for image-based vehicle classification and localisation. Generated dataset tackles some of the limitations of previous datasets and automates the labelled data production, often costly and time-consuming. The pipeline combines canvas generation and vehicle annotation based on position, scale, and colour alternations of seed image as well as prompt-based canvas outpainting to change the background context, season, and weather. Different metrics for generated image quality are utilised and augmented datasets are experimented in vehicle classification and localisation task with promising results.

**Strengths:**

Paper introduces important and timely topic of using recent generative machine learning tools for augmented data sampling and an example dataset. For the specific visual eye-level vehicle classification and localisation data, there exists few with some limitations, so there are needs for new easily produced datasets (and automated tools to generate the labelled examples). Paper is well-motivated for the application perspectives and gives appropriate related work of utilised tools and previous data in the focused domain. Usefulness of the data and pipeline is shown with quantitative metrics in relation to quality of generated images and usage of the images as augmented training data for the classifications outperforming the use of real data alone. Paper is clearly written and organized. Broader impact and limitations are discussed.

**Audience:**

Yes

**Broader Impact Concerns:**

Broader Impact Statement is present in the paper. Ethical and responsible use of the datasets/methodologies could be discussed more detailed, though.

**Claims And Evidence:**

Most of the paper claims are supported; 1) introducing the generated vehicle dataset and combination of existing methods to do it, 2) initial benchmark, showing the usefulness of augmented data in a focused classification task with improved results. What is a bit limited regarding the documentation of data organization, availability, maintenance, as well as responsible use. As stated above, what is a bit confusing is the relation between the proposed static dataset and the framework generating the data as the contributions. The generic framework, supporting the new dataset/example generation, could strengthen the paper and brought more value for end-users. From data quality point of view, more detailed analysis when the framework success and fails to produce proper images (and how different parameters affect) could better support the claims.

**Datasets And Benchmarks:**

Dataset collection/generation is documented mostly with sufficient details. There is a link to Github for codes and dataset. However, in the paper itself, there is no documented information about the organisation of the data, availability and maintenance, and ethical and responsible use (ethical issues very shortly mentioned in the Broader Impact section).

**Extended Submissions:**

To my knowledge, this is a new submission.

**Limitations:**

List of weaknesses and limitations
- Missing details of dataset (see Datasets and Benchmark section)
- Limited information about negative cases in the experiments
- Limited analysis of parameters and their effects to image quality and classification results
- Lack of promoting and publishing the generation framework for end-users
- Focusing only on generating very specific classes of images

**Requested Changes:**

There are several adjustments that needs to be done to consider work to be accepted:
1. As a dataset contribution, there should be more details of how the datasets would be published, accessed, and maintained in the future (hosting, licensing, maintenance plan)
2. Clearer description of the main output/contribution of the work. Is it "static" generated dataset and/or framework for end-users to generate new examples. In my opinion, publishing both the example dataset(s) and tools/framework that could be used for generating new labeled examples in vehicle classification or related scenarios would strengthen the work
3. There should be more detailed analyses/illustration of negative examples when the proposed generation pipeline fails (e.g., in contrast to Figure 3)
4. There should be more detailed analysis why certain parameters are chosen in the pipeline/methodologies, e.g., image quality assessment and how these affect the classification and localisation task

**Strengths And Weaknesses:**

Strengths
- Well-written and clearly organised manuscript
- Comprehensive motivation and related work section
- Experimental evaluation showing the usefulness of proposed approach

Weaknesses
- Missing details of dataset publishing, access, and maintenance plan
- Limited documentation and analysis why certain parameters in the pipeline are chosen; justification why these works (trial and error?)
- Limited information about negative cases, i.e., when the generation fails or produce low-quality images
- Unclear focus of producing new (benchmark/augmentation) dataset or tools/pipeline/framework for end-users to generate data
- Very focused target data and labels for eye-level vehicle images

---

### Review · Reviewer_WZKS · 2025-08-10

**Recommendation:** 2
**Confidence:** 3

**Summary Of Contributions:**

This paper introduces AIDOVECL, an AI-generated dataset of outpainted vehicles designed to address the scarcity of annotated eye-level vehicle images for classification and localization tasks. The dataset is created by detecting and cropping vehicles from manually selected seed images, which are then outpainted onto larger canvases using prompt-guided latent diffusion models (LDM) to simulate diverse real-world contexts. Automatic annotations (e.g., bounding boxes) are generated to provide high-quality ground truth. Experiments show that augmenting real datasets with AIDOVECL improves overall model performance by up to 8% and enhances predictions for underrepresented classes by up to 20%. The approach demonstrates the potential of outpainting as an automatic annotation paradigm, with applications in autonomous driving, urban planning, and environmental monitoring.

**Strengths:**

- Targets Critical Data Gaps: Directly addresses the lack of diverse, eye-level vehicle images in public datasets (e.g., Stanford Cars, COCO), which hinders applications like autonomous driving.

- Efficient Annotation: Automatically generates detailed bounding box annotations, drastically reducing manual labeling efforts.

- Proven Performance Boost: Augmenting real datasets with AIDOVECL enhances key metrics (e.g., mAP50, F1) and mitigates class imbalance effectively.

**Audience:**

Yes

**Broader Impact Concerns:**

The authors added relevant clarification to the original manuscript.

**Claims And Evidence:**

Yes.

**Datasets And Benchmarks:**

The authors provide links to open source repositories for the dataset and code.

**Extended Submissions:**

No.

**Limitations:**

- Inability to Generate Multi-Vehicle Scenes: The outpainting model fails to generate coherent scenes with multiple vehicles, limiting the dataset’s utility for training models to handle occlusions and complex real-world traffic scenarios.
- Reliance on Pretrained Models: The pipeline depends on pretrained detection (e.g., FCOS, RetinaNet) and outpainting models, which may fail to detect or realistically generate non-typical vehicles (e.g., rare truck models), introducing selection bias.
- Limited Seed Image Diversity: Seed images must be manually selected from "clean" sources meeting strict criteria (e.g., fully visible vehicles), restricting the diversity of generated outpainted images and hindering generalization.
- Class Confusion in Generated Images: Outpainting occasionally causes confusion between visually similar classes (e.g., coupes misclassified as sedans, pickups as SUVs) due to blurred masks or artificial feature extensions, reducing annotation accuracy.
- Restrictive Quality Assessment Metrics: The use of BRISQUE, CLIP-IQA, and TV loss may not fully capture perceptual realism, as these metrics do not explicitly evaluate contextual consistency (e.g., vehicle-environment lighting alignment).

**Requested Changes:**

**Reviewer Comments:**

- The inability to generate multi-vehicle scenes is a critical limitation, as occlusions and dense traffic are common in real-world scenarios. Please discuss technical barriers (e.g., model architecture, prompt design) preventing multi-vehicle generation and propose solutions, such as integrating spatial attention mechanisms to coordinate object placement.
- The reliance on pretrained detection models raises concerns about generalizability to rare or modified vehicles (e.g., custom trucks). It is recommended to evaluate detection performance on a diverse set of non-standard vehicles and explore fine-tuning detection models on underrepresented classes to reduce selection bias.
- The manual selection of seed images limits diversity. Please quantify the current seed image diversity (e.g., vehicle makes, models, conditions) and propose automated methods to expand the seed pool, such as using active learning to identify informative candidates from unlabeled data.
- Class confusion (e.g., coupes vs. sedans) undermines annotation reliability. Provide a detailed analysis of confusion patterns, including visual examples, and suggest refinements to the outpainting pipeline (e.g., tighter mask boundaries, class-specific prompts) to preserve distinguishing features.
- The quality assessment metrics (BRISQUE, CLIP-IQA, TV loss) do not explicitly measure contextual consistency (e.g., vehicle shadows matching scene lighting). It is advised to supplement these with human evaluation or metrics like LPIPS to better assess perceptual realism.
- The paper does not address potential overfitting to synthetic data. Please include ablation studies comparing model performance on synthetic-only vs. mixed (synthetic+real) training sets to verify that improvements stem from generalizable features, not synthetic artifacts.
- Background generation uses text-to-image models with negative prompts to exclude vehicles, but this may inadvertently remove relevant context (e.g., road signs). Evaluate the impact of background purity on model performance, and consider relaxing negative prompts to include contextually relevant non-vehicle objects.

**Strengths And Weaknesses:**

**Strengths:**

- Targets Critical Data Gaps: Directly addresses the lack of diverse, eye-level vehicle images in public datasets (e.g., Stanford Cars, COCO), which hinders applications like autonomous driving.

- Efficient Annotation: Automatically generates detailed bounding box annotations, drastically reducing manual labeling efforts.

- Proven Performance Boost: Augmenting real datasets with AIDOVECL enhances key metrics (e.g., mAP50, F1) and mitigates class imbalance effectively.

**Weaknesses:**
- Inability to Generate Multi-Vehicle Scenes: The outpainting model fails to generate coherent scenes with multiple vehicles, limiting the dataset’s utility for training models to handle occlusions and complex real-world traffic scenarios.
- Reliance on Pretrained Models: The pipeline depends on pretrained detection (e.g., FCOS, RetinaNet) and outpainting models, which may fail to detect or realistically generate non-typical vehicles (e.g., rare truck models), introducing selection bias.
- Limited Seed Image Diversity: Seed images must be manually selected from "clean" sources meeting strict criteria (e.g., fully visible vehicles), restricting the diversity of generated outpainted images and hindering generalization.
- Class Confusion in Generated Images: Outpainting occasionally causes confusion between visually similar classes (e.g., coupes misclassified as sedans, pickups as SUVs) due to blurred masks or artificial feature extensions, reducing annotation accuracy.
- Restrictive Quality Assessment Metrics: The use of BRISQUE, CLIP-IQA, and TV loss may not fully capture perceptual realism, as these metrics do not explicitly evaluate contextual consistency (e.g., vehicle-environment lighting alignment).